# Research Progress of Music Therapy on Gait Intervention in Patients with Parkinson’s Disease

**DOI:** 10.3390/ijerph19159568

**Published:** 2022-08-04

**Authors:** Zhuolin Wu, Lingyu Kong, Qiuxia Zhang

**Affiliations:** Physical Education and Sports School, Soochow University, Suzhou 215021, China

**Keywords:** Parkinson’s disease, music therapy, intervention therapy, gait, walking

## Abstract

Music therapy is an effective way to treat the gait disorders caused by Parkinson’s disease. Rhythm music stimulation, therapeutic singing, and therapeutic instrument performance are often used in clinical practice. The mechanisms of music therapy on the gait of patients with Parkinson’s disease include the compensation mechanism of cerebellum recruitment, rhythm entrainment, acceleration of motor learning, stimulation of neural coherence, and increase of cortical activity. All mechanisms work together to complete the intervention of music therapy on patients’ gait and help patients to recover better. In this paper, the effect of music therapy on gait disorders in Parkinson’s disease patients was reviewed, and some suggestions were put forward.

## 1. Introduction

Parkinson’s disease (PD) is a central nervous system disease characterized by low mortality and a high disability rate, with a prevalence rate of about 0.3% [1]. Older adults are at high risk for PD, with a prevalence rate of 1~2% in people over 65 and 3~5% in people over 85 [2]. Influenced by global ageing trends, the population of older persons with PD is projected to increase from 8.7 to 9.3 million by 2030 [3].

PD is mainly triggered by the depletion and loss of dopamine (DA) neurons in the substantia nigra pars compacta (SNpc) and the accumulation of cytoplasmic inclusion bodies in Lewy body (LB) neurons. PD patients usually have gait disorders and cognitive problems. Gait disorders can have a serious negative impact on their daily lives and are characterized by a shorter stride length, slower gait, and an unstable center of gravity [4], which are more pronounced in later stages of the disease [5]. Gait disorders may lead to daily mobility problems, increasing the risk of falls [6]. At present, drug therapy and deep brain stimulation are mainly used to relieve gait disorders in PD patients, but drug resistance in humans limits the effectiveness of drug therapy and side effects, such as abnormal limb movements, can affect the quality of daily life of patients [7]. Therefore, seeking an effective and sustained intervention to relieve gait disorders in PD patients is the focus of current clinical research.

Music is a sound wave composed of elements such as melody, rhythm, and harmony, and is defined as a melody consisting of a continuous combination of tones, arranged periodically [8]. Music activates the “auditory-motor pathway,” activating motor neurons through the rhythms contained in the music itself, causing muscles to contract and synchronizing body movements with the beat [9]. For example, music with a strong rhythm makes people want to be in sync with it, and the listener’s fingers or feet will automatically tap the beat. The main reason for this is that the auditory processing area of the cerebral cortex is adjacent to the motor area (MA). Due to the interaction between the auditory and motor systems, the human body will show corresponding motor performance when stimulated by music [10]. This motor response of the brain to auditory stimuli is called the thalamus response. Based on this principle, more and more scholars have applied music to clinical therapy, resulting in music therapy. MT plays a role through the cross-activity of the hypothalamic-pituitary-adrenal axis, nervous system, cardiovascular system, and digestive system to regulate the psychological and physiological functions of the human body, so it is widely used in the field of medicine and psychotherapy [11]. MT is also used as a special rehabilitation tool to intervene in the treatment of PD-induced gait disorders through a range of mechanisms, such as the cerebellum compensation mechanisms, rhythmic entrainment, accelerated motor learning, stimulation of neural coherence, and increased cortical activity. Based on relevant studies in recent years, this paper reviews the detailed mechanism of MT in the treatment of gait disorders in PD patients.

## 2. Forms of MT Conducted

MT can be divided into active and receptive treatments, depending on the treatment method. Active therapy involves patients playing musical instruments and participating in singing and opera performances under the guidance of a music therapist in a self-fulfilling form. It is a process involving all the sensory organs. The melody and rhythm of music are used as specific stimuli to obtain certain motor and emotional responses, thus combining the movement and stimulation of different sensory pathways, i.e., auditory and tactile (multisensory stimulation). Receptive therapy does not require the patient to actively participate in music performance or singing activities, just to listen to music [12]. Receptive music therapy is more significant in the treatment effect of PD patients with gait disorders, while active therapy is mainly in the form of teamwork, focusing more on the improvement of PD patients’ cognition, mood, quality of life, and other aspects.

## 3. Modalities and Clinical Outcomes of MT to Improve Gait in PD Patients

Current clinical treatment of gait disorders in PD patients consists mainly of Rhythmic Auditory Stimulation (RAS), Therapeutic Singing (TS), and Therapeutic Instrumental Music Playing (TIMP), treating patients by synchronizing or asynchronizing their walking rhythms with music to adjust walking parameters in this way [13].

### 3.1. RAS

RAS is a Neurological Music Therapy (NMT) technology that refers to auditory rhythm cues to promote the synchronization of gait movement [14] and is the main method for the intervention of gait disorders in PD patients. A large number of experiments have found that RAS can reshape sensorimotor rhythm and frontal-central parietal lobe/time connection and restore the brain mechanism that generates regular walking rhythm through the intervention of RAS in PD patients with gait disorders. In addition, other studies have shown that RAS has a good therapeutic effect on abnormal gait in PD patients. After treatment, bilateral ankle dorsiflexion and all gait parameters are significantly improved and the fall index decreases. Spatio-temporal gait parameters, such as stride speed and stride length, are significantly improved [15,16]. Simone [17] and McIntosh et al. [18] further found that PD patients walked synchronically according to the rhythm of music, and their step speed, stride frequency, and stride length were significantly improved, proving that MT combined with gait training could have a more obvious positive impact on the walking function of PD patients. In a study by Thaut et al. [19] on RAS training in PD patients, significant increases in gait speed and stride frequency were associated with significant changes in the timing of tibialis anterior and lateral femoral muscle electromyographic patterns. In addition, RAS can be combined with other training to achieve additional therapeutic benefits. Li et al. [20] used treadmill training combined with music intervention to intervene in the frozen gait of PD patients, and the research results showed that treadmill training combined with music intervention could significantly relieve the frozen gait of PD patients and reduce the number of falls. Therefore, RAS can be used not only as primary therapy but also as an adjunctive therapy.

### 3.2. TS

TS is a spontaneous behavior in which the patient actively completes the vocalization, an internal cue that differs from external cues, such as listening to rhythmic music. Patients can sing loudly or in their minds and synchronize their walk with the pace of the singing, which has large variability, at any time according to the adjustment of the state of patients, and the synchronized rhythm. Compared with external cues, singing does not require any equipment to play music, nor does it require the difficult task of consistently synchronizing with the music rhythm. It can be controlled according to the patient’s personal rhythm, which reduces spatial and temporal variability [21] and is easier for patients to complete. Therefore, TS has been regarded as an important method to improve gait disorders in PD patients in recent years. Elinor et al. [21] conducted an intervention study on 23 PD patients under concentrated conditions including singing and found that gait variability was reduced during singing, suggesting that TS is expected to be a novel and effective cue technology. Satoh et al. [22] had eight PD patients trained to perform walk exercises in sync with singing, and they showed signs of improvement in gait stability, limb coordination, and steering fluency after training. Harrison et al. [23] conducted a study of 60 patients with PD performing singing or singing in the mind, and it was found that relative to listening to the music rhythm, singing and singing in the mind have an obvious decrease in gait variability, and singing in the mind showed a more significant gait variability decrease. Therefore, singing loudly or singing in the mind can improve the gait disorders of PD patients to a certain extent, and this novel technique of using internal cues provides reliable guidance for PD rehabilitation treatment.

### 3.3. TIMP

TIMP has been used in several neurological disorders, including PD, in which patients actively play rhythmic melodies or beats to improve overall and fine motor skills through rhythmic entrainment. Playing musical instruments can provide immediate auditory feedback [24]. The instruments used here are mainly percussion instruments that can emphasize obvious beats, such as drums. Such instant auditory feedback can coordinate rhythm and movement, and percussive rhythm can stimulate the movement of patients to synchronize with it, and the movement of playing musical instruments can reduce the amplitude of involuntary movement of PD patients. TIMP is mostly used in group cooperation. Some people play musical instruments, while others walk and perform other body movements according to the rhythm of the music. This can not only treat movement disorders but also improve the cognitive and emotional quality of life. Pacchetti et al. [12] conducted an experiment on playing musical instruments for the treatment of PD, in which 16 patients with PD were selected to choose their favorite instruments for percussion or playing, while others walked to the rhythm. All patients received guidance from a professional music therapist prior to TIMP treatment, and formal treatment was started only after ensuring that the patient has a basic grasp of the essentials of playing an instrument. Patients were trained in TIMP intervention training once a week for three months. The results showed improved motor speed, improved motor retardation, improved communication and cooperation, and improved cognition. It is not just the interaction between PD patients, it is also the drumbeat of PD patients themselves used to synchronize walking. Pantelyat et al. [25] asked PD patients to beat West African drum rings to synchronize their movement with the rhythm of drum beating. All patients attended a West African drum circle class for training in West African drum circle playing before the intervention. After mastering the West African drum circle playing skills, patients underwent West African drum circle intervention training twice a week for six weeks. After the intervention, PD patients’ walking speed was improved to varying degrees, and their bradykinesia was effectively alleviated.

## 4. Study on Gait Mechanism of MT Intervention in PD Patients

Basal ganglia (BG) is generally considered to be the timing system of a beat [26] and is involved in both internal time and external rhythm perception [27]. However, the phosphorylation and profibrotic of α-synuclein in PD patients lead to the formation of LB and induced neuronal death, while LB is distributed in BG, which leads to the lesion and death of BG nerve cells, resulting in the impairment of BG function and the disturbance of time rhythm in patients, further resulting in time dysfunction [28]. It is the main reason for PD patients’ difficulty in perceiving music rhythm.

According to the MT brain stem reticular structure theory, musical rhythm stimulation activates BG in the brain stem reticular structure. The rhythm, melody, mode, and other factors in music are transmitted to the brain through the hearing cells, and the brain processes the different elements in different regions. The gait training exercise is performed under the cue of music, which requires coordination of the connection between movement and the auditory center, and the connection is the mechanism of MT acting on PD patients’ gait intervention. At present, there are many explanations for the gait intervention mechanism of MT in PD patients, which can be divided into the following: (1) compensation mechanism of cerebellum recruitment, (2) the underlying mechanism involves rhythm entrainment, (3) accelerated motor learning, (4) stimulating neural coherence, (5) increased cortical activity, etc. [29].

### 4.1. Compensation Mechanisms for Recruiting Cerebellum

Since BG is damaged in the brainstem of PD patients and the cerebellum is capable of processing temporal perception and motor execution, the temporal rhythmic perception of musical rhythms and performing synchronized movements are mainly accomplished by recruiting the cerebellum to compensate for the missing function. Braunlich et al. [30] investigated functional magnetic resonance imaging (fMRI) results and found that rhythmic motor behavior (right-hand finger tapping) performed by PD patients experimentally demonstrated increased functional neural connectivity between the auditory cortex and the executive control network, as well as between the executive control network and the cerebellum. The network of musically stimulated walking synchronized movements is the cerebellum-TC network (CTC), involved in the predictive coding of the temporal structure of movements and the matching of movements to exogenous cues [30].

The cerebellum is the central structure that controls internal time, in which the neo-cerebellum is able to process time and is also one of the clock mechanisms of the brain. The medial part of the cerebellum, the spinal cord, is mainly involved in sensory-motor processing, including motor timing. In contrast to the rhythm perception of the BG, the cerebellum performs more complex perceptual timekeeping processing, such as explicit and implicit timing [17]. Explicit timing refers to the need for explicit timing to deliberately estimate the duration and relies on an internal sense of time, while implicit timing refers to the use of external cues, relies less on conscious time-based judgments, and uses automatic timing systems [31]. Implicit and explicit timing have different underlying neural networks. Recessive timing is less dependent on BG and the supplementary motor area (SMA) [28], mainly through the compensation mechanism of recruiting the cerebellum. The cerebellum network relies on external cues (such as auditory or visual cues), and even if BG function is impaired, there will be no disturbance in the perception of time rhythm. It is generally not affected by PD [29]. Explicit timing recruits the BG, SMA, pre-motor cortex (PMC), and cerebellum [32]. However, the BG in PD patients is affected by LB and has impaired function, so it cannot perceive time rhythm normally. Calabro et al. [15] conducted gait training for PD patients under the intervention of RAS. After eight weeks of training, subjects showed improvements in the gait quality index. Their analysis of data recorded by electroencephalography showed that during RAS intervention walking, the body recruited parts of a network that included the cerebellum and different cortical areas. Although external cues such as implicit timing can avoid more complex time perception and avoid BG’s participation through the cerebellum’s compensation effect, there are also studies specifically on display timing, such as internal cues mainly focused on singing. Harrison et al. [23] had 30 PD patients complete walking exercises with internal and external cues. Results showed a decrease in gait variability for all subjects, and a decrease in gait variability for internal cues (singing) compared to external cues (RAS). This phenomenon may be because singing, as an internal cue for vocality-movement coupling, can better match one’s walking movements with one’s voice and adjust one’s walking movements at any time.

### 4.2. The Rhythm of Entrainment

The musical rhythm serves as an activation signal to stimulate the PD patient’s motor system to synchronize with the musical rhythm. In practice, this rhythmic entrainment synchronizes the walking movement with the musical rhythm and thus intervenes in the patient’s gait. “Entrainment” in physics refers to the locking of the frequencies of two moving oscillators in a stable rhythm or period, which have their frequencies or periods of movement when they move independently and a common period when they interact [30]. “Rhythm entrains” refers to the frequency at which the motion or signal frequency of one system entrains the other. However, the use of rhythmic music stimulation for PD patients is mainly to activate rhythm entrained, that is, the phase lock between auditory rhythm and body movement. This lock is the synchronization caused by rhythm entrained. The movement or signal frequency of one system induces another system to adopt the frequency synchronized with it. For example, rhythmic music is used to increase the intensity of spontaneous nerve oscillations in the motor center [33] and reconnect previously damaged motor pathways [34]. In the brain, the firing rates of auditory neurons triggered by auditory rhythms and music entrap the firing patterns of motor neurons, known as “motion-rhythm entrap” [35]. Within the cerebral cortex, rhythm processing and auditory–motor interactions occur in a widely distributed and layered neural network that extends from the brainstem and spinal cord levels to the cerebellum, basal ganglia, and cortical ring, thus allowing interactions between auditory and motor systems [36]. This motor response of the brain to auditory stimuli is called the thalamic response [10]. The motor response to the musical stimulus occurs simultaneously with the stimulus and is a spontaneous synchronization. The synchronous effect of rhythm entrainment is clinically applied in the gait rehabilitation of PD patients. Patients synchronize their movements with the rhythm by sensing the external time rhythm to achieve gait improvement. Motion-rhythm entraining relies on different brain regions, such as the auditory cortex, inferior parietal lobule, and frontal regions, such as SMA and PMC. Todd et al. [37] found that these regions seem to be unaffected by PD pathophysiology. McIntosh et al. [18] performed an intervention in 31 PD patients under different conditions and showed that the rhythmic music intervention group showed better improvements in mean walking step speed, step frequency, and stride length than the other groups. The authors suggested that this was due to the link between gait rhythm music training and auditory and motor centers. Therefore, the activation of the above network by rhythmic music is the main mechanism for improving dyskinesia in PD patients [38].

### 4.3. Accelerated Motor Learning

Music brings out different feelings in people. Often, listening to the right music during motor learning is more likely to increase people’s motor learning efficiency. This is because when people listen to the rhythm of music, the MA in the brain will become active, and the auditory environment and incentive effect of music may lead to the acceleration of motor learning [39]. The music signal is captured by a large number of hearing cells receiving a high frequency in the ear, and the signal is transmitted to the brain, regulating the brain nerve, and mobilizing a high and positive mental state, which will be conducive to the acceleration of motor learning. In walking exercises synchronized with rhythmic music in PD patients, the activation of the motor network associated with rhythmic perception will accelerate motor learning and increase the plasticity of learning by regulating the movement through the music and repeating the same movement each time. For example, the functional neural connections between the auditory cortex and the executive control network, as well as between the executive control network and the cerebellum, were increased when patients performed synchronous movement with rhythmic music stimulation [29]. The rich connection between the auditory and motor systems of the brain promotes the synchronization of walking and rhythmic music, and this connection has been described as a “backdoor” into the motor system and a means to improve the efficiency of motor learning [40]. Chuma et al. [41] experimented on motor learning of PD patients’ hand auditory cues, which showed that 12 PD patients showed higher training-induced plasticity after receiving the rhythmic music intervention compared with the control group. Music, as a sound cue, can promote the performance of various sports [42]. Synchronous walking with music can reduce the metabolic cost of exercise by improving neuromuscular or metabolic efficiency [43], increasing endurance, and reducing perceived fatigue. Therefore, this may be an important mechanism for disorders such as PD that affect motivation [1].

### 4.4. Stimulate Neural Coherence

When listening to music, music stimulates the nerve to produce excitement. These neural networks are coherent and orderly, and work together to play a role in the coordination of the human body’s motor system in the synchronous walking movement with rhythm music and intervene in gait. Braunlich et al. [42] studied the neural network recruitment of PD patients during rhythmic music cues, and fMRI showed that PD patients under rhythmic music showed stronger inter-network connections between auditory, executive control, and motor/cerebellar networks compared with healthy subjects. This is because the brain’s external rhythm stimulation (auditory) can drive more neurons to synchronize. When stimulated by rhythmic music, the auditory brain network (IC) extends to the bilateral auditory cortex. IC shows strong task-related activity, reflecting the amount of auditory stimulus. The network related to executive control is related to the posterior parietal lobe activity related to the dorsal attention network, and the bilateral lateral prefrontal lobe, inferior parietal lobe, and medial frontal lobe regions related to the executive control network are also studied by resting-state fMRI [30]. Motor/cerebellar networks include the motor and the somatosensory cortex [44]. The connections between these networks in PD patients may help compensate for the impaired corticostriatal motor control system [42]. Therefore, rhythmic music acts as a stimulant for neural coherence, stimulating neural coherence and enhancing the synchronization of neurons in the primary motor cortex [35].

### 4.5. Increased Cortical Activity

Perceiving musical rhythms is a process of perceiving time and walking in sync with it is a process of integrating the timing and sequence of movements that work through the relevant areas of the cerebral cortex. Rhythm music combines the sensory cortex and motor cortex regions. Grahn et al. [45] explored the role of BG in beat perception, and fMRI revealed that the auditory cortex, PMC, SMA, PMC, prefrontal cortex, pre-auxiliary motor area (pre-SMA), lateral cerebellum, and BG were activated and their interactions were enhanced.

During the synchronization of movements with rhythmic music, the brain needs to determine the order and time of each movement, which is a neural activity process that requires a high degree of integration of movement time and sequence. In terms of the sequence of actions, the frontal-parietal network consisting of the lateral prefrontal and inferior parietal areas, BG, and cerebellum were more active during actions. The perceptual-motor time was more closely related to SMA, right inferior frontal gyrus, central anterior sulcus, and bilateral superior temporal gyrus (STG). In complex rhythmic movements, such as discrete rhythmic movements, the cerebral cortex is divided into action sequences and perceptual movement time zones, and this pattern of partitioning is more pronounced than with sequential rhythmic movements [46]. When movements with sequential and temporal sequences are carried out, such as synchronous movement of walking and rhythmic music, temporal and sequential features are integrated into the brain’s zonal processing in the primary motor cortex, but independent in higher motor regions [47].

Due to the degeneration of DA neurons in the SNpc of PD patients and reduced DA content in the striatum (corpora striatum, CS), there is an association between DA system dysfunction and motor synchronization deficits [48]. The whole basal-ganglion-thalamic-cortical network is involved in the completion of actions and the perception of movement time [49]. Substantia nigra pars reticulate (SNr), the main output nucleus of BG, directly connects to the substantia nigra pars reticulate (SNr) through the hyper-direct pathway (HDP) that controls activation of the subthalamus nucleus (STN). It is an important structure in the human brain associated with PD and other involuntary movements [50]. Rhythmic music activated the superior temporal gyrus (STG), the SMA, the ventrolateral prefrontal cortex (VLPFC), and the MA involved in the perception of motor sequences. There is also some overlap in the cortical activation of the two, and the activation of the cortex will help the patient complete the rhythmic music motor synchronization training.

## 5. Conclusions

In conclusion, music therapy can effectively improve the effect of gait disorders in PD patients. However, there are still methodological problems in previous studies, such as small sample size and few objective physiological evaluation indicators. There are also still questions regarding the mechanism of action, such as which is the main mechanism of action under different stimulation conditions and whether the others are also involved in auditory and motor processing and learning.

To solve these problems, we can use more objective physiological indicators in future studies to assess the effectiveness of intervention effects, and it is also important to extend the follow-up period, including more age-diverse subjects, and exclude excessive interfering factors. Moreover, whether relevant elements of music, such as tonal audio or different types of music, can enhance specific brain regions should be studied, as well as further exploration of the mechanisms by which different types of music play a role in PD intervention.

## Data Availability

The data presented in this study are available upon request from the corresponding author.

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
