# Peer review of "Research Progress of Music Therapy on Gait Intervention in Patients with Parkinson’s Disease"

_ijerph, 2022, doi:10.3390/ijerph19159568_

Round 1
Reviewer 1 Report
Dear authors, congratulations on the choice of this very relevant topic.
The presented review is very interesting and can add very important knowledge, especially in the part on the physiological mechanisms in the nervous system with the application of music therapy in patients with Parkinson's disease, with the objective of improving gait.
The purpose of this review is presented in the abstract but must be included in the introduction.
The authors propose to review studies on the effect of music therapy on gait disorders in Parkinson’s disease patients. They show very important and little- discussed aspects of physiological mechanisms and indicators like compensation mechanism of cerebellum recruitment, rhythm entrainment, acceleration of motor learning, stimulation of neural coherence, and increase of cortical activity. However, it seems that the primary objective of this study would be to report the effects of music therapy on gait; so, the authors should explore and present in more detail the gait-related results of patients with Parkinson’s disease from intervention studies using music therapy.
It would be important to include the method section containing review type (narrative review?), consulted databases, exclusion criteria (Why were so many studies not included?). Which music therapy papers were considered relevant for the present review? How did you group the articles into these presented categories?
The arrangement of these categories and presentation of them is a little confusing, it is necessary to make a clearer order for readers, perhaps discuss physiological mechanisms first, then music therapy modalities and types of intervention.
The conclusion is a little vague, what is the contribution of this study from the clinical point of view in relation to the use of music therapy in the gait of patients with PD?
Reviewer 2 Report
- Shouldn't be mentioned in the introduction that many PD patients suffer from cognitive problems as well? I understand that the focus is on motor skills, but still ...
- Line 38-39: a combination of notes. This should - I think - be tones. Notes refer to musical notation. However, there are also aural traditions in which musical notation is not used. The word tones refers to sound.
- Line 56 and further: I would not speak of passive treatments, but of receptive treatments. Patients listen, and are very active in their listening. You write about things happening in the brain. Many brain areas show activity during a session in which patients listen.
- Lines 58-59: What is meant with a self-fulfilling form?
- You describe 3 kinds of music therapy. Isn't there a form of MT in which music and movement plays an important role, like the Dalcroze approach in education?
- Line 96: You describe TS as spontaneous behavior. I miss a little introduction to this form of MT. What makes it spontaneous? If a patient is taught to sing in order to get a grip on his or her movement, it's not spontaneous at all. I understand why singing can be beneficial, but I don't understand the spontaneous element in it.
- Lines 118-119, instant auditory feedback: does this require some sort of training/ learning? TIMP can be a powerful activity. It is however more complicated than it seems to be.
- Lines 136 and further: like you write in your conclusion: much more research should be done to draw robust conclusions on the interventions described. The subjects in many studies are rather small, the interventions short, with questionable effect sizes.
Overall, it's an interesting review study that should be the starting point for empirical studies.
